# Serum cobalamin in children with moderate acute malnutrition in Burkina Faso: Secondary analysis of a randomized trial

Henrik Friis[1]*, Bernardette Cichon[1], Christian Fabiansen[1,2], Ann-Sophie Iuel-Brockdorff[1], Charles W. Yaméogo[1,3], Christian Ritz[1], Ruth Frikke-Schmidt[4,5], André Briend[1,6], Kim F. Michaelsen[1], Vibeke B. Christensen[7,8], Suzanne Filteau[9ʘ], Mette F. Olsen[1,10ʘ]

1 Department of Nutrition, Exercise and Sports, University of Copenhagen, Frederiksberg, Denmark, 2 Paediatric Department, Holbæk Sygehus, Holbæk, Denmark, 3 Département Biomédical et Santé Publique, Institut de Recherche en Sciences de la Santé, Ouagadougou, Burkina Faso, 4 Department of Clinical Biochemistry, Rigshospitalet, Copenhagen, Denmark, 5 Department of Clinical Medicine, University of Copenhagen, Copenhagen, Denmark, 6 Center for Child Health Research, Tampere University, Tampere, Finland, 7 Médecins Sans Frontières–Denmark, Copenhagen, Denmark, 8 Department of Paediatrics, Rigshospitalet, Copenhagen, Denmark, 9 Faculty of Epidemiology and Population Health, London School of Hygiene & Tropical Medicine, London, United Kingdom, 10 Department of Infectious Diseases, Rigshospitalet, Copenhagen, Denmark

ʘ These authors contributed equally to this work.
* hfr@nexs.ku.dk

**Data Availability Statement:** As a university in a Member State of the European Union, University of Copenhagen is obliged to comply with the provisions of the General Data Protection

## Abstract

### Background

Among children with moderate acute malnutrition (MAM) the level of serum cobalamin (SC) and effect of food supplements are unknown. We aimed to assess prevalence and correlates of low SC in children with MAM, associations with hemoglobin and development, and effects of food supplements on SC.

### Methods and findings

A randomized 2 × 2 × 3 factorial trial was conducted in Burkina Faso. Children aged 6 to 23 months with MAM received 500 kcal/d as lipid-based nutrient supplement (LNS) or corn–soy blend (CSB), containing dehulled soy (DS) or soy isolate (SI) and 0%, 20%, or 50% of total protein from milk for 3 months. Randomization resulted in baseline equivalence between intervention groups. Data on hemoglobin and development were available at baseline. SC was available at baseline and after 3 and 6 months. SC was available from 1,192 (74.1%) of 1,609 children at baseline. The mean (±SD) age was 12.6 (±5.0) months, and 54% were females. Low mid-upper arm circumference (MUAC; <125 mm) was found in 80.4% (958) of the children and low weight-for-length z-score (WLZ; <−2) in 70.6% (841). Stunting was seen in 38.2% (456). Only 5.9% were not breastfed. Median (IQR) SC was 188 (137; 259) pmol/L. Two-thirds had SC ≤222 pmol/L, which was associated with lower hemoglobin. After age and sex adjustments, very low SC (<112 pmol/L) was associated

Regulation. Under Article 9 (2), (j) universities can process sensitive personal data for scientific research purposes. In addition, it is stipulated in Article 9, (4) that Member States may maintain or introduce further conditions, including limitations, with regard to the processing of genetic data, biometric data or data concerning health. The Danish legislation has introduced further conditions in Article 10 of the Danish Act on Data Protection. It is stated in Article 10 of the said Act, that personal research data can be transferred to scientific journals for verification of the research results. However, the Danish Act on Data Protection does not allow for personal data to be made available to others without prior individual approval from the Danish Data Protection Agency. Further information about the requirements of the said Article 10 is available on the University website https://informationssikkerhed.ku.dk/english/ protection-of-information-privacy/ academicpublications/. The Data Protection Officer of the University of Copenhagen can be contacted about data inquiries at dpo@adm.ku.dk.

**Funding:** The study was funded by Danish International Development Agency (09-097 LIFE) (KFM); Médecins Sans Frontières (Denmark, Norway) (VBC); Arvid Nilsson's Foundation (VBC); Merete and Mogens Brix Christensen (VBC); The World Food Program, which was part of a donation to the World Food Program from the American people through the support of the US Agency for International Development's Office of Food for Peace (VBC); the Alliance for International Medical Action; and the European Union's humanitarian aid funds, in partnership with Action Contre la Faim. The funders had no role in study design, data collection and analysis, decision to publish, or preparation of the manuscript.

**Competing interests:** I have read the journal's policy and the authors of this manuscript have the following competing interests: KFM has received research grants from US Dairy Export Council and the Danish Dairy Research Foundation, and also has research collaboration with Nutriset, a producer of LNS products, and patent owner; HF has received research grants from ARLA Food for Health Centre, and also has research collaboration with Nutriset, a producer of LNS products, and patent owner; AB was the inventor of LNS, for which Nutriset has the patent, but abandoned claims to royalties in 2003. Other authors declare no financial relationships with any organisations that might have an interest in the submitted work in the previous three years, and declare no other relationships or activities that could appear to have influenced the submitted work.

with 0.21 (95% CI: 0.01; 0.41, $p = 0.04$) and 0.24 (95% CI: 0.06; 0.42, $p = 0.01$) z-score lower fine and gross motor development, respectively.

SC data were available from 1,330 (85.9%) of 1,548 children followed up after 3 months and 398 (26.5%) of the 1,503 children after 6 months. Based on tobit regression, accounting for left censored data, and adjustments for correlates of missing data, the mean (95% CI) increments in SC from baseline to the 3- and 6-month follow-up were 72 (65; 79, $p < 0.001$) and 26 (16; 37, $p < 0.001$) pmol/L, respectively. The changes were similar among the 310 children with SC data at all 3 time points. Yet, the increase was 39 (20; 57, $p < 0.001$) pmol/L larger in children given LNS compared to CSB if based on SI (interaction, $p < 0.001$). No effect of milk was found. Four children died, and no child developed an allergic reaction to supplements. The main limitation of this study was that only SC was available as a marker of status and was missing from a quarter of the children.

## Conclusions

Low SC is prevalent among children with MAM and may contribute to impaired erythropoiesis and child development. The SC increase during supplementation was inadequate. The bioavailability and adequacy of cobalamin in food supplements should be reconsidered.

## Trial registration

ISRCTN Registry ISRCTN42569496.

## Author summary

### Why was this study done?

- We searched PubMed since 2000 using the terms (severe malnutrition OR acute malnutrition OR marasmus OR kwashiorkor OR wasting) AND (cobalamin OR B12) AND child*).

- We found no studies among children with moderate acute malnutrition (MAM) or severe acute malnutrition (SAM) treated with recommended food aid products and with data on serum cobalamin (SC) before and after treatment.

### What did the researchers do and find?

- We found that two-thirds of children with MAM had marginal or low SC, which were associated with deficits in hemoglobin and child development scores.

- The content of cobalamin in the food aid products was in accordance with the World Health Organization (WHO)'s recommendation and the duration of supplementation, 3 months, much longer than in programs. Despite this, one-third still had marginal or low SC at the end of supplementation, and mean levels declined considerable in the subsequent 3 months.

**Abbreviations:** AGP, α₁-acid glycoprotein; CRP, C-reactive protein; CSB, corn–soy blend; DS, dehulled soy; FAO, Food and Agriculture Organization; IF, intrinsic factor; IGF-1, insulin-like growth factor 1; LAZ, length-for-age z-score; LNS, lipid-based nutrient supplement; MAM, moderate acute malnutrition; MDAT, Malawi Development Assessment Tool; MUAC, mid-upper arm circumference; NHANES, National Health and Nutrition Examination Survey; RUTF, ready-to-use therapeutic food; SAM, severe acute malnutrition; SC, serum cobalamin; SI, soy isolate; WHO, World Health Organization; WLZ, weight-for-length z-score.

- While the study showed that supplementation was inadequate in terms of normalizing cobalamin status, it showed that lipid-based nutrient supplement (LNS) based on soy isolate (SI) was the best.

## What do these findings mean?

- Cobalamin is important for erythropoiesis and brain development. It is also likely to be of pivotal importance in the development of malnutrition, given its role in enterocyte turnover and, hence, absorption.
- Millions of children are treated for acute malnutrition every year. It is important that these children not only survive, but also get enough cobalamin to support key functions and long-term development.

## Introduction

In the first few years of life, growth velocity is high, and the brain reaches close to adult size. Hence, the nutritional requirements are high, and inadequate dietary intake may have long-lasting effects on growth and development. The diet of children in low-income settings is based on a starch-rich staple, with limited amounts of fruits, vegetables, and pulses, and little animal source foods [1]. Lack of animal source foods contribute to a range of coexisting deficiencies [1].

Exposure to pathogens and mycotoxins may impair intestinal absorption and nutritional status [2]. On a background of food insecurity, these factors may contribute to the widespread micronutrient deficiencies. Research and programs have focused on iron, zinc, and vitamin A deficiency. Cobalamin (B-12) deficiency has received less attention. Since cobalamin is important for DNA synthesis, developing deficiencies will first impair rapidly dividing cell lines such as enterocytes and erythrocytes [3]. Cobalamin is also involved in sprouting and myelination of neurons and affects brain development [3]. Cobalamin is solely found in animal source or fortified foods, and emerging data confirm that low or marginal cobalamin status is widespread, especially in Asia and Africa [4].

We conducted a randomized food supplementation trial among young children with moderate acute malnutrition (MAM) in Burkina Faso and used a factorial design to assess the effects of food type, soy quality, and milk content on the primary outcome fat-free mass index [5] and on the key secondary outcomes: iron status [6] and child development [7]. We have previously reported that food supplements were beneficial for child development, which motivated us to assess the role of cobalamin.

The aim of this paper is 3-fold. First, to assess the level and correlates of serum cobalamin (SC) at baseline, i.e., before initiation of supplementation. Second, to assess the associations of low SC with hemoglobin and child development at baseline. Third, during 3 months food supplementation, to assess the effects of matrix, soy quality, and milk content on SC.

## Methods

### Study design, area, and participants

The study was based on data from Treatfood (ISRCTN42569496), a randomized 2 × 2 × 3 factorial trial assessing the effects of food supplements among children with MAM [5] in Province

du Passoré, Northern Region, Burkina Faso, in 2013 to 2014. Children in the area were screened by community health workers based on mid-upper arm circumference (MUAC) or by screening teams based on both MUAC and weight-for-length z-score (WLZ). Children could also be taken to the site by the caregiver or be referred from a health center. The final assessment for eligibility was carried out by study staff at the sites. Children between 6 and 24 months who were residents in the catchment area were recruited if a diagnosis of MAM was confirmed, i.e., MUAC 115 to <125 mm and/or WLZ −3 to <−2 in the absence of edema. Children were excluded if treated for severe acute malnutrition (SAM; i.e., WLZ <−3 or MUAC <115 mm or edema) or hospitalized within the past 2 months, if already in a nutritional program, or requiring hospitalization, e.g., hemoglobin <50 g/l. Children with a severe disability, limiting the feasibility of investigations, or with suspected allergy to milk, peanuts, corn–soy blend (CSB), or lipid-based nutrient supplement (LNS) were also excluded. In order to reduce mixing or sharing of the supplements, we only included the first child with MAM identified from a household, but provided the same supplement to siblings with MAM, as well as to discordant twins. During the intervention period, children visited the health center every 2 weeks. Serious adverse events were defined as anaphylaxis and death.

## Study intervention

The children were randomized to receive a daily supplement for 3 months, with either a LNS or a CSB, and with either dehulled soy (DS) or soy isolate (SI), and either 0%, 20%, or 50% of protein from dried skimmed milk. LNS was provided in sachets of 92 g per daily ration and CSB in 1.7-kg bags per 14-day ration. All supplements consisted of 500 kcal/daily serving (120 g CSB or 92 g LNS). LNS was ready to use, whereas CSB needed to be mixed with water and made into porridge. Supplements were manufactured by GC Rieber Compact A/S (Søfteland, Norway). The nutrient composition of the products complied with the World Health Organization (WHO)'s technical note for management of MAM [8]. The content of water-soluble vitamins was higher in CSB to account for degradation during cooking. The cobalamin content per daily ration from premix was 4.1 μg in CSB and 3.2 μg in LNS. The dried skimmed milk contained additional cobalamin, so that the cobalamin content in products with 20% and 50% of protein from milk was 9.5% and 23.8% higher than products without milk.

As described previously [5,7], we individually randomized children to 1 of 12 combinations of the 3 factors, with variable block size and stratified by site, using www.randomization.com by a person not otherwise involved in the trial. The supplements were designated by a 1-letter code by the manufacturer, which was placed in a 12-letter sequence labeled on each supplement. The code key was kept in a sealed envelope in a safe until study completion. Only one person, otherwise not involved in the study, was aware of the random sequence and code system and relabeled supplements with individual study identifications. Study participants, outcome assessors, and other study staff were blinded with respect to soy quality, milk content, and matrix, while it was not possible to blind study participants with respect to matrix.

## Data collection

At baseline, study nurses collected information about sociodemographics, 2-week morbidity, and carried out clinical examination. Children who were not up to date with vaccinations were referred to a health center. Children received albendazole (200 mg if <8 kg body weight; 400 mg if >8 kg body weight) and vitamin A (100,000 IU if 4 to 8 kg body weight; 200,000 IU if >8 kg body weight) if they had not received a supplement in the previous 6 months. Weight was measured to the nearest 100 g using an electronic scale with double weighing function (Seca model 881, Hamburg, Germany). Length was measured to the nearest 1 mm with a wooden

length board. WLZ was determined at sites using WHO simplified field tables [9] and later recalculated using the zscore06 package [10] in Stata 14 (Stata, College Station, Texas, US). MUAC was measured on the left arm to the nearest 1 mm using a standard measuring tape. Anthropometric measurements were taken in duplicate by trained staff. Child development was assessed using an adapted version of the Malawi Development Assessment Tool (MDAT) [11]. Adaptation, validation, and pilot testing of the MDAT tool has been reported previously [12]. In brief, we assessed the gross motor, fine motor, and language domains of the MDAT. Items were rated as passed (1 point) or failed (0 points) until the child had failed 6 consecutive items. The assessor would then mark the remaining items as failed and move to the next domain. We calculated MDAT z-scores from a Malawian population of healthy nonmalnourished children [11].

Dietary data were collected using qualitative 24-hour recall. A list of 25 food groups was based on internationally available questionnaires from WHO [13], Food and Agriculture Organization (FAO) [14], as well as research about diets in Ouagadougou carried out by the Institut de Recherche pour le Développement [15] and adapted to the local context. The 25 food groups were then aggregated into 7 food groups as suggested by WHO [13].

Venous blood was collected from the arm at baseline and after 3 and 6 months. One drop was used for diagnosis of malaria (*Plasmodium falciparum*) using a rapid diagnostic test (SD Bioline Malaria Ag P.f), and 1 drop was used to measure hemoglobin on site using a HemoCue device (Hb 301, Ängelholm, Sweden). Remaining blood was put into a sample tube with clot activator (BD reference #368,492) and transported to the trial laboratory at 2 to 8°C. Serum was isolated following centrifugation at 700 g for 5 minutes (EBA 20S Hettich) and stored at −20°C until shipment and then at −80°C until analysis. Serum C-reactive protein (CRP), $\alpha_1$-acid glycoprotein (AGP), and ferritin were determined at VitMin Lab in Willstaedt, Germany using a combined sandwich enzyme-linked immunosorbent assay [16]. All samples were measured in duplicate and the intra- and interassay coefficients of variation were <10%. Serum ferritin was adjusted for inflammation using regression models (15). The cutoffs used was 12 and 24 μg/L for serum ferritin, 2 and 10 mg/L for serum CRP, and 0.8 and 1.2 g/L for serum AGP.

SC was determined at Rigshospitalet, Denmark, using standard hospital assays on a Cobas 8000, e801 module fulfilling accreditation from the Danish Accreditation Fund. The quantification method was competitive electrochemiluminescence immunoassay, and the measuring range was 112 to 1,480 pmol/L. Coefficient of variation was 6% at both 220 pmol/L (control H1. Immunoassay Plus Control Liquicheck Bio-Rad Kat. 361, Human serum) and 550 pmol/L (control H3. Immunoassay Plus Control Liquicheck Bio-Rad Kat. 363, Human serum). SC was considered low if ≤148 pmol/L and marginal if between 149 and 222 pmol/L, to allow comparison with data from surveys published by Allen and colleagues [4]. The detection limit of 112 pmol/L defined very low SC.

## Statistical analyses

As previously reported, the sample size was calculated for the primary outcome of the Treatfood trial [5]. With a sample size of 1,608 (i.e., 134 in each of the 12 combinations resulting from the $2 \times 2 \times 3$ factorial design), we had 80% power to detect a 0.6 SD difference in any pairwise comparison with 5% significance level, while allowing for 20% loss to follow-up. If no interaction, the sample size would allow detection of smaller effects.

Data were double entered into Epidata 3.1 software (Epidata Association, Odense, Denmark) and double entry checks were carried out on a daily basis. All statistical analyses were

carried out using Stata 14. SC was not a preplanned outcome of the trial; however, we specified which covariates to assess as potential correlates and confounders before we received the cobalamin data and performed the statistical analyses.

**Baseline.** Tobit regression was used to assess for potential correlates of baseline SC, while adjusting for age and sex and taking the detection limit of 112 pmol/L into account as left censoring [17]. Linear regression was used to assess the associations between baseline SC and baseline hemoglobin and MDAT z-scores, while adjusting for age, sex, serum ferritin (hemoglobin only), and elevated serum CRP and AGP.

**Follow-up.** Mean SC at 0, 3, and 6 months and changes from 0 to 3 and 6 months were estimated using mixed-effects tobit regression with normally distributed random effects to describe the variation between children. Adjustments were made for age, sex, month of admission, site, and baseline MUAC, WLZ, length-for-age z-score (LAZ), as well as other correlates of missing cobalamin data. Mixed-effects tobit regression was also used to evaluate the effect of matrix, soy quality, and amount of milk on SC. First, 3-way interactions between the factors were tested for using likelihood ratio tests and, where possible, reduced to 2-way interactions or main effects. Analyses were adjusted for baseline SC as a covariate, and age, sex, MUAC, WLZ, LAZ, site, and elevated serum CRP and morbidity. Model checking was based on residual and normal probability plots. The change in SC during and after supplementation were presented as estimated means with 95% CIs. $p$-Values <0.05 were considered significant.

## Ethical considerations

All children in need received treatment free of charge according to an adapted version of the Integrated Management of Childhood Illnesses guidelines [18,19] and the national protocol. Children who developed SAM during the intervention period were treated with ready-to-use therapeutic food (RUTF; Plumpy'Nut, Nutriset, Malaunay, France). Children who did not recover from MAM subsequently received RUTF or were referred for medical investigation. The study was carried out in accordance with the Declaration of Helsinki. Consent was obtained from caregivers, prior to inclusion, verbally and in writing (signature or fingerprints). Data were kept confidential and in a locked facility. The Ethics Committee for Health Research of the Government of Burkina Faso (2012-8-059) approved the study, and the Danish National Committee on Biomedical Research Ethics (1,208,204) gave consultative approval. The trial was registered as ISRCTN42569496.

## Results

Data on SC at baseline were available from 1,192 (74.1%) of 1,609 children in the Treatfood trial conducted from September 2013 to August 2014. SC data were also available from 1,330 (85.9%) of the 1,548 children followed up after 3 months and 398 (26.5%) of the 1,503 children followed up after 6 months (**Fig 1**). As reported previously, 4 children died, and no child developed an allergic reaction to supplements [5].

Among the 1,192 with baseline SC data, mean (±SD) age was 12.6 (±5.0) months, and 54% were females. Low MUAC (<125 mm) was found in 80.4% (958) of the children and low WLZ (<−2) in 70.6% (841). Stunting was seen in 38.2% (456). Only 5.9% were not breastfed. The 417 children without baseline SC were younger (11.7 versus 12.6 months, $p = 0.001$), had higher serum CRP (2.8 versus 2.2 mg/L, $p = 0.03$) and AGP (1.5 versus 1.3 g/L, $p < 0.001$), and were more likely to be breastfed (96.6 versus 94.0%, $p = 0.04$), but there were no differences with respect to sex distribution, anthropometry, morbidity, hemoglobin, and measures of child development (**Table 1**). There were no differences between children with and without SC data after 3 months. After 6 months, children with no data on SC were older and had

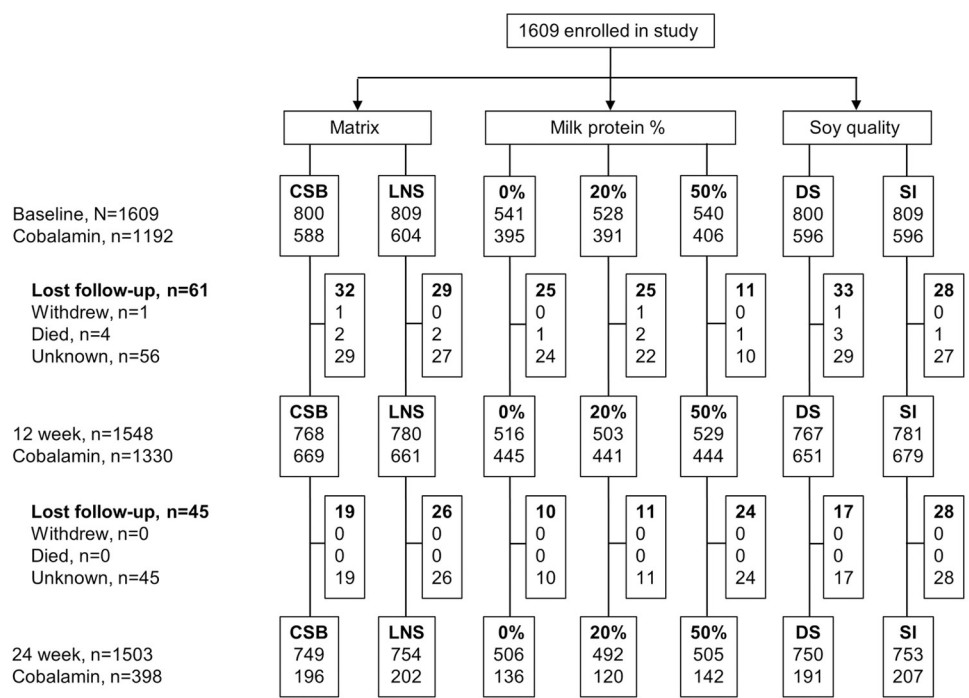

**Fig 1. Trial profile.** Of 1,609 children enrolled, cobalamin data were available on 1,192 at baseline. After 12 and 24 weeks, 1,548 and 1,503 children were followed up, and cobalamin data were available on 1,330 and 398, respectively. CSB, corn–soy blend; DS, dehulled soy; LNS, lipid-based nutrient supplement; SI, soy isolate.

**Table 1. Baseline characteristics of 1,609 children with MAM with or without cobalamin data at baseline and 3- and 6-month follow-up.**

| | Baseline | | | 3-month follow-up | | | 6-month follow-up | | |
|---|---|---|---|---|---|---|---|---|---|
| | Data $n$ = 1,192 | No data $n$ = 417 | $p$-Value | Data $n$ = 1,330 | No data $n$ = 279 | $p$-Value | Data $n$ = 398 | No data $n$ = 1,211 | $p$-Value |
| **Sociodemographic characteristics** | | | | | | | | | |
| Age, months | 12.6 (±5.0) | 11.7 (±4.4) | 0.001 | 12.4 (±4.9) | 12.0 (±4.6) | 0.18 | 11.8 (±4.7) | 12.5 (±4.9) | 0.006 |
| Sex, female | 54.1% (645) | 56.3% (234) | 0.48 | 53.8 (716) | 58.4 (163) | 0.16 | 51.8% (206) | 55.6% (673) | 0.19 |
| **Breastfed** | 94.0% (1,119) | 96.6% (402) | 0.04 | 94.4% (1,254) | 96.0% (267) | 0.26 | 94.5% (376) | 94.7% (1,145) | 0.86 |
| **Anthropometry** | | | | | | | | | |
| MUAC, mm | 122.7 (±4.0) | 122.5 (±3.9) | 0.34 | 122.6 (±3.9) | 122.5 (±4.2) | 0.54 | 123.0 (±4.1) | 122.5 (±3.9) | 0.051 |
| WLZ | −2.2 (±0.5) | −2.2 (±0.5) | 0.16 | −2.2 (±0.5) | −2.3 (±0.5) | 0.20 | −2.2 (±0.5) | −2.2 (±0.5) | 0.21 |
| LAZ | −1.7 (±1.1) | −1.7 (±1.1) | 0.88 | −1.7 (±1.1) | −1.7 (±1.2) | 0.77 | −1.7 (±1.2) | −1.7 (±1.1) | 0.42 |
| **Morbidity** | | | | | | | | | |
| Ill last 2 weeks | 37.6% (445) | 39.2% (163) | 0.56 | 38.8% (513) | 34.2% (95) | 0.15 | 46.9% (186) | 35.1% (422) | <0.001 |
| Serum CRP, mg/L | 2.2 (0.7; 9.2) | 2.8 (1.0; 10.5) | 0.03 | 2.4 (0.8; 9.8) | 2.2 (0.7; 8.5) | 0.41 | 3.3 (0.8; 11.5) | 2.1 (0.7; 8.6) | 0.01 |
| Serum AGP, g/L | 1.3 (±0.6) | 1.5 (±0.7) | <0.001 | 1.4 (±0.7) | 1.3 (±0.7) | 0.66 | 1.4 (±0.7) | 1.3 (±0.6) | 0.003 |
| **Hemoglobin, g/L** | 100 (±16) | 100 (±17) | 0.69 | 100 (±16) | 101 (±16) | 0.13 | 97 (±15) | 101 (±16) | <0.001 |

Data shown as mean (±SD), median [IQR], or $n$ (%).

AGP, $\alpha_1$-acid glycoprotein; CRP, C-reactive protein; LAZ, length-for-age z-score; MAM, moderate acute malnutrition; MUAC, mid-upper arm circumference; WLZ, weight-for-length z-score.

higher serum concentrations of inflammatory markers and hemoglobin ($p < 0.05$, **Table 1**). As previously reported, randomization resulted in baseline equivalence between intervention groups [7]. Missing SC was not associated with any of the 3 interventions neither at baseline, 3 months nor 6 months (all $p > 0.18$, **Fig 1**).

## Level and correlates of baseline serum cobalamin

The median (IQR) SC at baseline was 188 (137; 259) pmol/L and ≤222 pmol/L in 63.4%. SC was lower in the youngest children (**Table 2**). After adjustment for age and sex, there were no

**Table 2. Growth, inflammation, and morbidity as correlates of baseline SC (pmol/L) in 1,192 children with MAM[1].**

|  |  | Unadjusted | | Age–sex adjusted | |
|---|---|---|---|---|---|
|  | *n* | B (95% CI) | *p* | B (95% CI) | *p* |
| Sex |  |  |  |  |  |
| Female | 645 | - |  |  |  |
| Male | 547 | −8 (−22; 6) | 0.27 | −11 (−25; 2) | 0.10 |
| Age (month) |  |  |  |  |  |
| 6 to 11 | 629 | - |  | - |  |
| 12 to 18 | 339 | 26 (11; 42) | 0.001 | 28 (12; 44) | 0.001 |
| 18 to 24 | 224 | 52 (34; 70) | <0.001 | 53 (35; 71) | <0.001 |
| **Anthropometry** |  |  |  |  |  |
| MUAC (mm) |  |  |  |  |  |
| >125 | 234 | - |  |  |  |
| ≤125 | 958 | 3 (−15; 20) | 0.75 | 4 (−14; 22) | 0.65 |
| WLZ |  |  |  |  |  |
| >−2 | 351 | - |  |  |  |
| ≤−2 | 841 | 8 (−7; 23) | 0.29 | 5 (−11; 20) | 0.55 |
| LAZ |  |  |  |  |  |
| >−2 | 736 | - |  |  |  |
| >−3 and ≤−2 | 334 | 10 (−6; 25) | 0.23 | 1 (−14; 17) | 0.85 |
| ≤−3 | 122 | −10 (−33; 13) | 0.40 | −23 (−46; 1) | 0.06 |
| **Inflammation and morbidity** |  |  |  |  |  |
| Serum CRP (mg/L) |  |  |  |  |  |
| ≤2 | 569 | - |  |  |  |
| >2 and ≤10 | 341 | 6 (−10; 22) | 0.46 | 2 (−14; 18) | 0.78 |
| >10 | 282 | −6 (−24; 11) | 0.47 | −9 (−26; 8) | 0.31 |
| Serum AGP (g/l) |  |  |  |  |  |
| ≤0.8 | 241 | - |  | - |  |
| >0.8 and ≤1.2 | 366 | 11 (−8; 31) | 0.26 | 7 (−12; 27) | 0.45 |
| >1.2 | 585 | 15 (−4; 33) | 0.11 | 7 (−11; 25) | 0.46 |
| Ill last 2 weeks |  |  |  |  |  |
| No | 739 | - |  | - |  |
| Yes | 445 | −6 (−20; 9) | 0.43 | −8 (−22; 6) | 0.25 |
| Malaria (Rapid test) |  |  |  |  |  |
| Negative | 726 | - |  | - |  |
| Positive | 463 | −16 (−30; −2) | 0.03 | −21 (−35; −6) | 0.004 |

[1]Data shown as number (*n*), regression coefficient B (95% CI), and *p*-value. Numbers in categories may not add up, due to missing values.

AGP, $\alpha_1$-acid glycoprotein; CRP, C-reactive protein; LAZ, length-for-age z-score; MAM, moderate acute malnutrition; MUAC, mid-upper arm circumference; SC, serum cobalamin; WLZ, weight-for-length z-score.

Table 3. Breastfeeding and food groups as correlates of baseline SC (pmol/L) in 1,192 children with MAM[1].

| | | Unadjusted | | Age–sex adjusted | |
|---|---|---|---|---|---|
| | *n* | B (95% CI) | *p* | B (95% CI) | *p* |
| **Breastfeeding and food intake**[2] | | | | | |
| Breastfed currently | | | | | |
| Yes | 1,119 | - | | - | |
| No | 72 | 107 (79; 135) | <0.001 | 89 (60; 119) | <0.001 |
| Grains, roots, and tubers | | | | | |
| No | 178 | - | | - | |
| Yes | 1,009 | 24 (4; 43) | 0.02 | 9 (−11; 29) | 0.39 |
| Legumes and nuts | | | | | |
| No | 1,014 | - | | - | |
| Yes | 178 | 14 (−4; 34) | 0.13 | 6 (−14; 25) | 0.57 |
| Vitamin A–rich fruits and vegetables | | | | | |
| No | 857 | - | | - | |
| Yes | 334 | 8 (−7; 24) | 0.29 | 1 (−15; 16) | 0.92 |
| Other fruits and vegetables | | | | | |
| No | 529 | - | | - | |
| Yes | 663 | 13 (−1; 27) | 0.06 | 1 (−14; 15) | 0.94 |
| Dairy | | | | | |
| No | 1,147 | - | | - | |
| Yes | 45 | 40 (3; 76) | 0.03 | 38 (3; 74) | 0.03 |
| Meat | | | | | |
| No | 1,117 | - | | - | |
| Yes | 75 | 15 (−14; 43) | 0.32 | 9 (−20; 37) | 0.60 |

[1]Data shown as number (*n*), regression coefficient B (95% CI), and *p*-value. Numbers in categories may not add up, due to missing values.

[2]Dietary data based on 24-hour recall.

MAM, moderate acute malnutrition; SC, serum cobalamin.

differences by category of WLZ or MUAC. Children with LAZ <−3, i.e., severe stunting, had 23 (95% CI: −1; 46) pmol/L lower levels, although not significant ($p = 0.06$). Positive malaria test was associated with 21 (95% CI: 6; 35, $p = 0.004$) pmol/L lower levels, whereas elevated markers of inflammation were not.

Not currently breastfed was associated with 89 (95% CI: 60; 119, $p < 0.001$) pmol/L higher SC (**Table 3**). Only 3.8% had dairy, and 6.3% had meat the previous day. Yet, intake of dairy was associated with 38 (95% CI: 3; 74, $p = 0.03$) pmol/L higher SC, whereas intake of meat or any other food groups was not ($p > 0.05$). If assessed in the same model, dairy intake was associated with a 30 (95% CI: −5; 66, $p = 0.09$) pmol/L higher and not currently breastfed with 87 (95% CI: 57; 116, $p < 0.001$) pmol/L higher SC.

## Serum cobalamin and hemoglobin

SC <112 pmol/L was associated with 5.7 (95% CI: 3.0; 8.3, $p < 0.001$) g/L lower hemoglobin (**Table 4**), and levels 112–148 and 149–222 pmol/L with 1.8 (95% CI: -0.6; 4.1, $p = 0.15$) and 2.3 (95% CI: 0.3; 4.3, $p = 0.02$) g/L lower hemoglobin after adjustment for other known correlates.

## Serum cobalamin and child development

SC <112 pmol/L was associated with 0.21 (95% CI: 0.01; 0.41, $p = 0.04$) lower fine motor and 0.24 (95% CI: 0.06; 0.42, $p = 0.01$) lower gross motor z-score (**Table 5**). Further adjustment for

**Table 4. SC and other correlates of baseline hemoglobin (g/L) in 1,192 children with MAM[1].**

| | n | Age–sex adjusted | | Multivariable model | |
|---|---|---|---|---|---|
| | | B (95% CI) | p | B (95% CI) | p |
| Sex | | | | | |
| Female | 645 | - | | | |
| Male | 547 | −3.9 (−5.7; −2.2) | <0.001 | −3.4 (−5.0; −1.7) | <0.001 |
| Age (month) | | | | | |
| 6 to 12 | 629 | - | | | |
| 12 to 18 | 339 | −3.8 (−5.8; −1.7) | <0.001 | −2.6 (−4.6; −0.6) | 0.01 |
| 18 to 24 | 224 | −4.4 (−6.7; −2.0) | <0.001 | −3.3 (−5.6; −1.1) | 0.004 |
| **Inflammation** | | | | | |
| Serum CRP (mg/L) | | | | | |
| ≤2 | 569 | - | | - | |
| >2 and ≤10 | 341 | −6.6 (−8.6; −4.6) | <0.001 | −4.6 (−6.7; −2.4) | <0.001 |
| >10 | 282 | −10.6 (−12.7; −8.6) | <0.001 | −7.4 (−10.0; −4.9) | <0.001 |
| Serum AGP (g/L) | | | | | |
| ≤0.8 | 241 | - | | - | |
| >0.8 and ≤1.2 | 366 | −4.0 (−6.4; −1.7) | 0.001 | −2.9 (−5.4; −0.5) | 0.02 |
| >1.2 | 585 | −10.2 (−12.5; −8.0) | <0.001 | −6.2 (−8.8; −3.5) | <0.001 |
| **Micronutrient status** | | | | | |
| Serum ferritin (adj, μg/L) | | | | | |
| ≤12 | 462 | −3.7 (−5.8; −1.6) | <0.001 | −4.3 (−6.3; −2.3) | <0.001 |
| >12 and ≤24 | 341 | −0.3 (−2.5; 2.0) | 0.81 | 0.1 (−2.0; 2.2) | 0.95 |
| >24 | 384 | - | | | |
| SC (pmol/L) | | | | | |
| <112 | 159 | −5.5 (−8.3; −2.8) | <0.001 | −5.7 (−8.3; −3.0) | <0.001 |
| ≥112 and ≤148 | 212 | −1.7 (−4.2; 0.8) | 0.18 | −1.8 (−4.1; 0.6) | 0.15 |
| >148 and ≤222 | 387 | −2.4 (−4.5; −0.3) | 0.03 | −2.3 (−4.3; −0.3) | 0.02 |
| >222 | 437 | - | | - | |

[1]Data shown as numbers (n), regression coefficients B (95% CI), and p-value. Numbers in categories may not add up, due to missing values. The multivariable model contains all variables shown.

AGP, α$_1$-acid glycoprotein; CRP, C-reactive protein; MAM, moderate acute malnutrition; SC, serum cobalamin.

elevated levels of markers of inflammation did not change the associations, although it became not significant for fine motor z-score (−0.19, 95% CI: −0.39; 0.003, p = 0.054). SC between either 112 to 148 or 149 to 222 pmol/L was not associated with fine or gross motor z-scores. SC was not associated with language z-score.

## Effects of interventions

During food supplementation, the median (IQR) SC increased from 188 (137; 259) to 255 (188; 344) pmol/L, at which time 38.0% still had values ≤222 pmol/L. After an additional 3 months without supplementation, median SC dropped to 206 (154; 271) pmol/L. The medians were similar among 310 children with SC data at all 3 time points, i.e., 187 (137; 258), 250 (178; 336), and 215 (160; 275) pmol/L at baseline, 3 and 6 months, respectively. This was consistent with findings from a tobit regression model with adjustment for age, sex, site, month of admission, and baseline MUAC, WLZ, LAZ, as well as inflammation, morbidity, and hemoglobin, found to be correlates of missing cobalamin data at 6 months (**Table 1**). Based on this model, the mean (95% CI) SC at baseline, 3 and 6 months were 205 (198; 212), 277 (270; 283),

**Table 5. SC as correlate of MDAT z-scores in 1,192 children with MAM[1].**

| | | Language domain | | Fine motor domain | | Gross motor domain | |
|---|---|---|---|---|---|---|---|
| | | Age–sex | Multivariable | Age–sex | Multivariable | Age–sex | Multivariable |
| | $n$ | B (95% CI) | B (95% CI) | B (95% CI) | B (95% CI) | B (95% CI) | B (95% CI) |
| SC (pmol/L) | | | | | | | |
| <112 | 159 | −0.07 (−0.26; 0.12) $p = 0.47$ | −0.05 (−0.23; 0.13) $p = 58$ | −0.21 (−0.41; −0.01) $p = 0.04$ | −0.19 (−0.39; 0.003) $p = 0.054$ | −0.24 (−0.42; −0.06) $p = 0.01$ | −0.23 (−0.47; −0.05) $p = 0.01$ |
| ≥112 and ≤148 | 212 | 0.003 (−0.17; 0.17) $p = 0.98$ | 0.02 (−0.15; 0.19) $p = 83$ | 0.07 (−0.11; 0.25) $p = 0.44$ | 0.08 (−0.10; 0.26) $p = 0.38$ | −0.06 (−0.22; 0.11) $p = 0.52$ | −0.05 (−0.22; 0.12) $p = 0.56$ |
| >148 and ≤222 | 387 | 0.01 (−0.13; 0.15) $p = 0.90$ | 0.02 (−0.12; 0.16) $p = 0.81$ | 0.10 (−0.05; 0.25) $p = 0.21$ | 0.11 (−0.04; 0.26) $p = 0.17$ | −0.08 (−0.22; 0.06) $p = 0.26$ | −0.08 (−0.22; 0.06) $p = 0.24$ |
| >222 | 437 | - | - | - | - | - | - |

[1]Data shown as number ($n$), regression coefficient B (95% CI), and $p$-value. Numbers in categories may not sum up, due to missing values. In the multivariable model further adjustments are made for elevated serum CRP and AGP.

AGP, $\alpha_1$-acid glycoprotein; CRP, C-reactive protein; MDAT, Malawi Development Assessment Tool; MUAC, mid-upper arm circumference; SC, serum cobalamin.

and 231 (220; 241) pmol/L, respectively. The mean (95% CI) increments in SC from baseline to the 3- and 6-month follow-up were 72 (65; 79, $p < 0.001$) and 26 (16; 37, $p < 0.001$) pmol/L, respectively (**Fig 2**). The means were similar among 310 children with SC determined at all 3 time points (S1 Fig).

The effects of food matrix, soy quality, and milk content on SC after 3 months of supplementation are shown in **Table 6**. There was a 3-way interaction between the experimental factors ($p = 0.02$) explained by an interaction between matrix and soy quality ($p < 0.001$), due to a larger increase in SC in those given LNS (versus CSB) if SI was used. The main effect of LNS (versus CSB) of 16 (95% CI: 3; 29, $p = 0.02$) pmol/L was larger with SI (39, 95% CI: 20; 57, $p < 0.001$) than with DS (−8, 95% CI: −26; 11, $p = 0.42$). There were no effects of milk content.

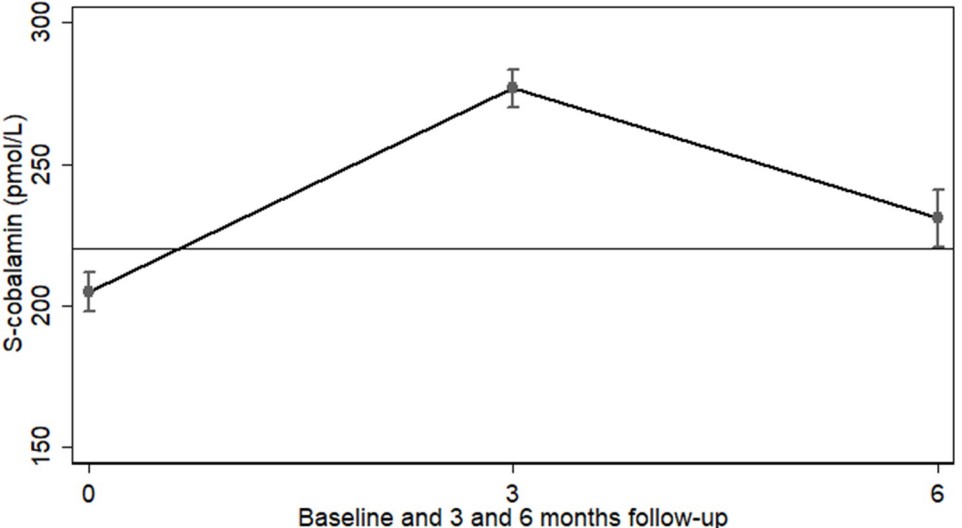

**Fig 2. SC during and after supplementation.** Mean (95% CI) SC at baseline ($n = 1,192$), after 3 months supplementation ($n = 1,330$), and after an additional 3 months without supplementation ($n = 398$). Based on mixed-effects tobit regression and adjusted for age, sex, months of admission, site and baseline values of MUAC, WLZ, and LAZ, serum CRP, inflammation and morbidity, and baseline SC. Horizontal line at 222 pmol/L indicates cutoff to define normal SC. CRP, C-reactive protein; LAZ, length-for-age z-score; MUAC, mid-upper arm circumference; SC, serum cobalamin; WLZ, weight-for-length z-score.

**Table 6. Effects of matrix, soy quality, and milk content in supplementary foods on SC (pmol/L) at end of the 3 months supplementation among 1,016 children with MAM[1].**

| | Main effects | | | Effect of LNS versus CSB by soy quality | | | | | Effects of SI versus DS by matrix | | | | |
|---|---|---|---|---|---|---|---|---|---|---|---|---|---|
| | | | | SI | | | DS | | | LNS | | CSB | |
| **Matrix** | | | | | | | | | | | | | |
| LNS | 16[2] | (3; 29) | 0.02 | 39 (20; 57) | | <0.001 | −8 (−26; 11) | | 0.42 | | | | |
| CSB | - | - | | - | | | - | | | | | | |
| **Soy quality** | | | | | | | | | | | | | |
| SI | −11[2] | (−24; 2) | 0.10 | | | | | | | 11 (−6; 30) | 0.27 | −35 (−53; −16) | <0.001 |
| DS | - | - | | | | | | | | - | | - | |
| **Milk content** (% of protein) | | | | | | | | | | | | | |
| 50% | 9 | (−7; 25) | 0.30 | | | | | | | | | | |
| 20% | −2 | (−18; 14) | 0.84 | | | | | | | | | | |
| 0% | - | - | | | | | | | | | | | |

[1]Data shown as mean difference (95% CI) based on intention-to-treat population. Tobit regression adjusted for baseline measure of the outcome, WLZ, LAZ, MUAC, age, sex, month of admission, and site.

[2]Interaction between matrix and soy quality: $p < 0.001$.

CSB, corn–soy blend; DS, dehulled soy; LAZ, length-for-age z-score; LNS, lipid-based nutrient supplement; MAM, moderate acute malnutrition; MUAC, mid-upper arm circumference; SC, serum cobalamin; SI, soy isolate; WLZ, weight-for-length z-score.

## Discussion

SC was low in children with MAM in Burkina Faso, as two-thirds had marginal or low levels. This is expected in low-income populations with a low intake of animal source and fortified foods and a high burden of gastrointestinal infections. Children will be born with low body stores of cobalamin, receive breastmilk with low content, and subsequently be given complementary foods with little if any animal source or cobalamin-fortified foods. A recent review summarized prevalence data from nationally representative studies and large nonrepresentative studies [4]. Of 6 studies among children in Asia and Africa, the prevalence of low and marginal cobalamin combined, i.e., below 222 pmol/L, ranged from to 33% to 80%.

Our study was conducted among children with MAM, diagnosed by low MUAC and/or low WLZ. Within these ranges of MUAC and WLZ, there were no association with SC but severe stunting was associated with lower SC. Although we are unable to make causal inferences based on cross-sectional data, this association could be due to a diet with little animal source foods being a cause of both cobalamin deficiency as well as impaired linear growth. Animal source foods are the only sources of natural cobalamin, but also an excellent source of growth nutrients, such as zinc, phosphorus, and amino acids, which are essential to synthesis of lean body mass and to linear growth [1]. Although cobalamin is essential to purine and pyrimidine metabolism and hence to DNA synthesis, it is not considered a growth nutrient [20]. However, a trial among young children in India found that cobalamin supplementation resulted in a 0.07 increase in weight-for-age z-score [21]. While this was due to an increase in ponderal, but not linear growth, cobalamin supplementation increased linear growth in subgroups of children who were either stunted or wasted. It is also possible that the association between impaired linear growth and cobalamin in our study could be explained by environmental enteric dysfunction [2]. This multifactorial condition is characterized by increased permeability of the gut which leads to systemic inflammation and down-regulation of insulin-like growth factor 1 (IGF-1), an important growth factor. It is also accompanied by impaired absorptive capacity, which would lead to deficiencies of growth nutrients, hence contributing

to impaired growth. It would also lead to cobalamin deficiency, which may impair regeneration of enterocytes and thereby further contribute to impaired absorptive capacity.

In contrast to other markers of micronutrient status, SC was not associated with elevated serum levels of the acute phase reactants CRP and AGP. While this is in accordance with findings from the US National Health and Nutrition Examination Survey (NHANES) study among adults [22], as well as nationally representative surveys among women and children [23], prospective studies are needed to confirm if indeed the validity of SC is unaffected by inflammation. Yet, a positive malaria antigen test was associated with reduced SC. If this reflects that malaria infection impairs cobalamin status through a mechanism independent of the inflammatory response, then it might be through uptake of cobalamin from the blood by the plasmodium parasite, which also has cobalamin-dependent methionine synthase [24]. Interestingly, it has been reported that the fish tapeworm (*Diphyllobothrium latum*) may cause megaloblastic anemia by digesting cobalamin in the human small intestine [25,26].

Only around 5% of the children had been fed dairy or meat the previous day, which means that the majority of children get dairy or meat less then weekly, if ever. While intake of animal source foods is likely to be low among the general population of children in the area, it is most probably lower in these children with MAM and may have contributed to its development. Intake of dairy, but not meat, was associated with higher cobalamin. We neither had data on type of dairy, i.e., local or commercial cow's milk, cheese, etc., nor on amount or frequency of intake. A study among 2-year-old Norwegian children assessed the correlation between dietary intake and SC [27]. Despite better dietary data, based on 7-day food records, and much larger intakes of animal source foods, only intake of dairy was correlated with SC.

In our study, children who were not breastfed had higher SC compared to those being breastfed, even after adjustment for age and sex. From a large study among 6 to 30 months old north Indian children, a similar difference in SC between breastfed and nonbreastfed children (183 versus 334 pmol/L) was reported. Dietary data were not available, but intake of animal source or fortified foods was considered low [28]. In our study, further adjustment for intake of animal source foods did not change the association. This may well be due to residual confounding, since we only had crude data on intake from food groups.

## Serum cobalamin and hemoglobin

Despite limitations of a cross-sectional study, the considerably lower hemoglobin in children with marginal or low SC probably reflects that cobalamin deficiency impairs erythropoiesis. Cobalamin and folate play important roles in pyrimidine and purine metabolism and are therefore essential to DNA synthesis and cell division [3]. Since the turnover of blood cells is rapid, these cells are affected early in the development of cobalamin deficiency. In contrast to folate, which is found in a large range of foods, including green leafy vegetables and legumes, cobalamin is only available from animal source or fortified foods. Since the intake of animal source foods is low in most low-income settings, it is likely that cobalamin deficiency is among the most widespread deficiencies and may contribute considerably to anemia.

## Serum cobalamin and child development

Suboptimal SC was associated with lower fine and gross motor domain z-scores, but not with language domain z-scores. Deficits were only seen with very low SC (<112 pmol/L). This is in contrast to hemoglobin, for which lower levels were seen even in children with marginal SC. A prospective study among 12- to 18-month-old children in India showed that plasma cobalamin was not associated with mental development index score at baseline, but positively associated with mental development index score after 4 months [29]. A later factorial trial found that

cobalamin supplements increased gross motor development, whereas cobalamin and folic acid combined also increased problem-solving [30].

## Effects of interventions

The lack of effects of milk, the only animal source ingredient in the foods, might be due to its modest contribution to cobalamin intake compared to the premix. Addition of dried skimmed milk to achieve 20% and 50% of protein from milk only resulted in 9.5% and 23.8% more cobalamin compared to the 4.1-μg cobalamin in CSB products and 3.2 μg in the LNS products without milk, which only contained cobalamin from premix. Interestingly, although the daily ration of CSB contained more cobalamin, those given LNS (versus CSB) had a greater increase in SC, but only if the LNS contained SI. A higher increase in those given LNS could be explained by a greater degradation during preparation of porridge from CSB than accounted for by increasing the content. It could also be due to a relatively greater consumption of the LNS compared to CSB rations [31,32]. Finally, it could be due to a greater bioavailability of cobalamin from the LNS matrix. The latter explanation is more likely given the interaction with soy quality. We have previously reported the same interaction between matrix and soy quality in relation to weight gain from this trial, i.e., children receiving LNS had a greater weight gain than those receiving CSB, but only if the products contained SI [5].

The overall increase in SC during supplementation was insufficient as more than one-third of the children were left with marginal or low SC. Furthermore, during the subsequent 3 months, there was a considerable decline reflecting low intake and continued diversion of cobalamin to target tissues. Our data show that the best supplement with respect to repletion of cobalamin status is LNS with SI. However, the content of cobalamin in LNS should be reconsidered, and cobalamin bioavailability in children with MAM should be studied. Cobalamin is primarily absorbed through a specific, saturable system involving binding to an intrinsic factor (IF) released in the stomach and uptake through IF receptors in the ileum [3]. An additional 1% to 2% of cobalamin in the diet can be absorbed through the unspecific system, and this is particularly important if IF-mediated absorption is impaired or saturated [3]. Both specific and unspecific absorption is likely to be affected, but does not seem to have been studied. Low cobalamin status and inadequate repletion during treatment are likely bigger problems in children with SAM, known to have reduced gastric acidity accompanied by bacterial overgrowth [33] and reduced enterocyte mass [34].

## Strengths and limitations

The main strength is the large sample size, giving precision and power, and data on dietary intake, hemoglobin and development, as well as the randomized trial allowing us to assess effects of the experimental factors on SC. The findings are likely to be generalizable to children with MAM in low-income settings in sub-Saharan Africa, with limited access to animal source and fortified foods.

The main limitation is missing data on SC, mainly due to lack of serum. For the longitudinal data, missing data were associated with age, morbidity, inflammation, and hemoglobin, which may indicate that data were missing at random (i.e., missing data might be predicted by observed data). Consequently, analyses using mixed-effects models will still provide unbiased results. The observation is further corroborated by results from analyses that included adjustment for correlates of missingness and, also, complete case analyses only including children with data at all 3 time points; all these analyses provided very similar findings. For the cross-sectional analyses, missing values were also associated with age and inflammation, and, therefore, only results from analyses adjusted for correlates of missing were reported.

Another limitation is the cross-sectional design used to assess associations between SC and hemoglobin and child development. However, it is well established that adequate cobalamin intake and status is critically important for hemopoiesis and child development. The aim of assessing these associations was therefore merely to corroborate SC cutoffs, based on these known cobalamin-dependent outcomes, and to assess to what extent low cobalamin in the population is associated with functional deficits. While the associations we find between low SC and both low hemoglobin and low child development scores are likely to reflect the well-known cause–effect relationship, it cannot be inferred from the cross-sectional design.

The lack of data on serum methylmalonic acid, the only specific marker of cobalamin deficiency, is not necessarily a limitation with respect to the analysis on hemoglobin, since the aim was to assess associations between categories of SC and known cobalamin-dependent outcomes. It is likely that the role of cobalamin for these outcomes is through the essentiality of methylcobalamin for de novo synthesis of purine and pyrimidine, rather than through the essentiality of deoxyadenosylcobalamin as a cofactor for methylmalonyl-CoA mutase. If, furthermore, erythropoiesis and early child development rank higher in the hierarchy of biological functions requiring cobalamin than the functions dependent on methylmalonyl-CoA mutase, then impaired erythropoiesis and child development may develop before any increase in serum methylmalonic acid.

## Conclusions

Children with MAM have low SC, which may contribute to impaired erythropoiesis and child development. Food products used for treatment fail to replete cobalamin status. The bioavailability and adequacy of cobalamin in food supplements for children with acute malnutrition should be reconsidered.

## Supporting information

**S1 Fig. SC during and after supplementation.** Mean (95% CI) SC at baseline, after 3 months supplementation, and after an additional 3 months without supplementation among 398 children with SC data at all time points. Based on tobit regression and adjusted for age, sex, months of admission, site and MUAC, WLZ and LAZ, inflammation and morbidity. Horizontal line at 222 pmol/L indicates cutoff to define normal SC. LAZ, length-for-age z-score; MUAC, mid-upper arm circumference; SC, serum cobalamin; WLZ, weight-for-length z-score.
(TIF)

**S1 Text. STROBE checklist.** STROBE, Strengthening the Reporting of Observational Studies in Epidemiology.
(DOCX)

**S2 Text. CONSORT checklist.** CONSORT, Consolidated Standards of Reporting Trials.
(DOCX)

## Author Contributions

**Conceptualization:** Henrik Friis, André Briend, Kim F. Michaelsen, Vibeke B. Christensen, Suzanne Filteau.

**Data curation:** Henrik Friis, Bernardette Cichon, Christian Fabiansen, Ann-Sophie Iuel-Brockdorff, Charles W. Yaméogo.

**Formal analysis:** Henrik Friis, Christian Ritz, Mette F. Olsen.

**Funding acquisition:** Henrik Friis, Kim F. Michaelsen, Vibeke B. Christensen.

**Methodology:** Henrik Friis, Christian Ritz, Ruth Frikke-Schmidt, André Briend, Kim F. Michaelsen, Suzanne Filteau, Mette F. Olsen.

**Validation:** Mette F. Olsen.

**Writing – original draft:** Henrik Friis.

**Writing – review & editing:** Henrik Friis, Bernardette Cichon, Christian Fabiansen, Ann-Sophie Iuel-Brockdorff, Charles W. Yaméogo, Ruth Frikke-Schmidt, André Briend, Kim F. Michaelsen, Vibeke B. Christensen, Suzanne Filteau, Mette F. Olsen.

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
