## [Editor Report · Decision Letter 0]

9 Jul 2021

Dear Dr Friis, 

Thank you for submitting your manuscript entitled "Serum cobalamin in children with moderate acute malnutrition: findings from a randomized trial in Burkina Faso" for consideration by PLOS Medicine.

Your manuscript has now been evaluated by the PLOS Medicine editorial staff as well as by an academic editor with relevant expertise and I am writing to let you know that we would like to send your submission out for external peer review.

Kind regards,

Beryne Odeny

Associate Editor

PLOS Medicine

---

## [Decision Letter · Decision Letter 1]

8 Oct 2021

Dear Dr. Friis,

Thank you very much for submitting your manuscript "Serum cobalamin in children with moderate acute malnutrition: findings from a randomized trial in Burkina Faso" (PMEDICINE-D-21-02962R1) for consideration at PLOS Medicine. 

[LINK]

In light of these reviews, I am afraid that we will not be able to accept the manuscript for publication in the journal in its current form, but we would like to consider a revised version that addresses the reviewers' and editors' comments. Obviously we cannot make any decision about publication until we have seen the revised manuscript and your response, and we plan to seek re-review by one or more of the reviewers. 

We expect to receive your revised manuscript by Oct 29 2021 11:59PM. Please email us (plosmedicine@plos.org) if you have any questions or concerns.

We look forward to receiving your revised manuscript. 

Sincerely,

Beryne Odeny, 

PLOS Medicine

plosmedicine.org

1) Please revise your title according to PLOS Medicine's style. Your title must be nondeclarative and not a question. It should begin with main concept if possible. For example, please place the study design in the subtitle (i.e., after a colon). For example, “Serum cobalamin in children with moderate acute malnutrition in Burkina Faso: A cross-sectional study of findings from a randomized trial” 

2) Please delete the “RESEARCH IN CONTEXT” section – this is not required by PLOS Medicine

3) At this stage, we ask that you write a non-technical Author Summary. The Author Summary should immediately follow the Abstract in your revised manuscript. This text is subject to editorial change and should be distinct from the scientific abstract. The summary should be accessible to a wide audience that includes both scientists and non-scientists. Please see our author guidelines for more information: https://journals.plos.org/plosmedicine/s/revising-your-manuscript#loc-author-summary.

4) Abstract:

a) Please report your abstract according to CONSORT for abstracts: http://www.consort-statement.org/extensions?ContentWidgetId=562 . Structure your abstract using the PLOS Medicine headings (Background, Methods and Findings, Conclusions).

b) Please combine the Methods and Findings sections into one section, “Methods and findings”. Replace “Interpretation” with “Conclusions”

c) Please ensure that all numbers presented in the abstract are present and identical to numbers presented in the main manuscript text.

d) Please quantify all results (with p values in addition to 95% CI).

e) Please include a summary of adverse events if these were assessed in the study.

f) In the last sentence of the Abstract Methods and Findings section, please describe the main limitation(s) of the study's methodology.

5) Did your study have a prospective protocol or analysis plan? Please state this (either way) early in the Methods section:

6) Please ensure that the study is reported according to the STROBE guideline for observational studies, and include the completed STROBE checklist as Supporting Information. Please add the following statement, or similar, to the Methods: "This study is reported as per the Strengthening the Reporting of Observational Studies in Epidemiology (STROBE) guideline (S1 Checklist)." The STROBE guideline can be found here: http://www.equator-network.org/reporting-guidelines/strobe/

7) Your study is observational and therefore causality cannot be inferred. Please remove language that implies causality throughout the manuscript, such as “effect/s.” Refer to associations instead. 

8) In the Methods and Results section, please consistently provide 95% CIs and p values for estimates in the main text and tables.

9) In table 5 and 6, please provide the meaning of the bolded numbers in the footnotes.

10) Please provide the meaning of bars and whiskers in your figures.

11) Please replace the terms “girls” with “females,” and “boys” with “males”

12) Please replace the term “marginally significant” with “not significant”

13) Please remove the ‘Funding’, from the end of the main text. In the event of publication, this information will be published as metadata based on your responses to the submission form.

14) Please use the "PLOS Medicine" citation style for reference formatting and see our website for other reference guidelines https://journals.plos.org/plosmedicine/s/submission-guidelines#loc-references. 

a) Please ensure that only six names precede et al.

b) Remove the “Public Library of Science” from ref #7

Comments from the reviewers:

Reviewer #1: This is an important report of secondary data analysis of an RCT with results published in 2017 in this journal. There are other analyses from this same data (well cited in this manuscript) however the authors do not tell us relevant findings from those in the background. Here the authors look at the i) level and the determinants/correlates of cobalamin serum levels (and haemoglobin and child development) at the baseline of trial, and ii) the effects of food supplementation (matrix, soy quality and milk content) on serum cobalamin during the follow-up. The analysis is not conducted as in an RCT. Interaction coefficients were decided upon a p-value calculation. It is a well-written report however there are a few issues to be addressed before publication.

Major issue:

- I have to commend the authors to conduct a Tobit regression for the multivariable analysis of the serum cobalamin to address the fact that measurement is left censored. However, some of the analysis was conducted with the replacement of these censored values with 112/√2. This is an arbitrary choice. Did you conduct sensitivity analysis with another value? Why not use Tobit when comparing this serum cobalamin in all analyses rather than the t-tests or ANOVA?

Minor issues:

- Line 145 better to write weeks rather than wk

- Line 151 please cite the zscore06 package and tell us the version of these WHO reference tables

- Line 151 cite properly Stata 12; and line 189 Stata 14.

- Line 188 cite properly the Epidata 3.1 software

- Line 158 please add a justification for using Malawian MDAT z-scores for Burkina Faso children

- Line 199 please write "linear mixed-effects models" rather than "xtmixed models". Xtmixed is a Stata command therefore only a few Stata users would recognise it.

- Line 233 I think it would be better if it is written "level and correlates of baseline serum cobalamin"

- There is an important amount of missing data on cobalamin. For the longitudinal data analysis, the linear mixed-effects analysis implicitly deals with missing data under Missing at Random assumption (I think that is fine). At the baseline, the randomization may have offered the MCAR (missing completely at - random) so it is correct to ignore the missed measurements (although table 1 offers some concerning p-values at baseline). I think it would be good to discuss the data missing in strengths/limitations.

Reviewer #2: 

Thank you very much for asking me to look at this work. In general, this is a very well written manuscript and follows on several papers that have been written from the clinical trial

There are a number of minor issues worthy of comment here:

1. The paper combines two study designs: a cross sectional study and a randomized controlled trial. Clearly a cross sectional study has its own limitations since the exposure and outcome are being are studied at the same time.

2. The second issue is that of the sample size for the study. This section is missing in this manuscript. The basis for the sample size including the effect size, power and precision is missing. 

3. While the objectives were clearly stated, the RCT should have had a clear hypothesis regarding the intervention and cobalamin etc.

4. The process of randomization was not adequately described in the manuscript. This includes the process itself, allocation concealment etc. It is possible that the process was described in more detail elsewhere in the papers preceding this one, so the authors could describe it to a certain extent and refer to the detailed description in the previous papers from the same work.

5. Why were cobalamin results available for only 1192/1609 children? Despite this comment, the authors have clearly compared those with the data and those without. This is commendable.

6. The issue of other infections other than bacteria were not considered in detail. Is it possible that other infections such as Ascaris lumbricoides, hookworm and HIV affected the baseline levels? We note that the children were give albendazole at the beginning of the study. Be that as it may, it would appear that this is an area this is an area needing further study.

7. There is need for study profile

8. The authors have published part of this work [1] but without looking at cobalamin. In that paper the authors assessed gross motor, fine motor, language and cognitive skills. It is not clear why the current paper has not looked at cognitive function and cobalamin deficiency.

References

1. Olsen MF, Iuel-Brockdorff AS, Yaméogo CW, Cichon B, Fabiansen C, et al. (2020) Impact of food supplements on early child development in children with moderate acute malnutrition: A randomised 2 x 2 x 3 factorial trial in Burkina Faso. PLOS Medicine 17(12): e1003442. https://doi.org/10.1371/journal.pmed.1003442

Reviewer #3: General

The objective of this study was to describe the prevalence of low vitamin B12 status (serum cobalamin, SC) among Burkinabé children 6-23 mo of age with moderate acute malnutrition (MAM) participating in a food supplementation trial, and examine the impact of various food supplements on SC. As noted by the authors, there is a general lack of information on B12 status among children in Africa, and particularly among children with MAM. This paper thus adds new information on vitamin B12 deficiency and the effects of food supplements on micronutrient status in this population. In general the paper is well written and methods are appropriate.

There was a relatively high proportion of missing baseline SC values (26%). The authors' analysis of characteristics of children with available vs missing samples provides insight on the possible direction of bias. However, I do not see a comparison of whether/how missing samples differed by intervention group. This would be helpful to report as differential missing values by group could potentially influence the conclusions on the intervention effects.

The abstract concludes that "The SC increase during supplementation was inadequate. The bioavailability and adequacy of cobalamin in food supplements should be reconsidered." And in Lines 361-364: "The overall increase in serum cobalamin during supplementation was insufficient as more than one-third of the children were left with marginal or low serum cobalamin. Furthermore, during the subsequent 3 months there was a considerable decline. This suggests that the bioavailability of cobalamin is compromised in children with MAM."

The observation that a large proportion still had low SC after 3 months of treatment is important and understanding the role of bioavailability would be useful. I wonder if another explanation (just speculation) might be that B12 is diverted quickly to functional sites (including bone marrow) during repletion so that less remains in circulation compared to a person with adequate B12 status.

However, I disagree that the interpretation of the decrease after 3 months of no supplementation signals a bioavailability problem. This decrease is not surprising as SC concentrations are responsive to recent intake and would be expected to decline if children discontinue supplemental food are returning to a diet that is very low in animal-sourced foods or other sources of B12. 

Minor

Lines 126-128: Any relevant exclusion criteria?

Lines 140-141: "The cobalamin content from premix was 4.1 μg in CSB and 3.2 μg in LNS." Is this the content per recommended serving?

The detection limit of 112 pmol/L seems somewhat high compared with the cutoff for deficiency <149 pmol/L.

Methods: What year(s) was the study carried out vs what year was SC measured? That is, how long were samples stored, and under what conditions, prior to analysis? This would be especially useful since the outcome is not specifically listed in the trial registration.

Lines 194-196 and 199-200 and 206-207: How was multicollinearity assessed in the models (especially if multiple anthropometric measures were used in the same model?)

Minor note: Line 200 refers to WHZ while line 206 refers to WLZ. 

Lines 303-305: For associations between B12 and inflammation, please see this recent paper by the BRINDA group (Young et al, AJCN 2020). This study also found no association in multiple cross-sectional surveys.

Lines 353-355: What information is available on the stability of B12 in LNS or CSB? Are there data to suggest that stability is lower in CSB?

Reviewer #4: See attachment

Michael Dewey

[LINK]

---

## [Decision Letter · Decision Letter 2]

26 Jan 2022

Dear Dr. Friis,

Thank you very much for re-submitting your manuscript "Serum cobalamin in children with moderate acute malnutrition: findings from a randomized trial in Burkina Faso" (PMEDICINE-D-21-02962R2) for review by PLOS Medicine.

I have discussed the paper with my colleagues and the academic editor and it was also seen again by two reviewers. I am pleased to say that provided the remaining editorial and production issues are dealt with we are planning to accept the paper for publication in the journal.

[LINK]

We look forward to receiving the revised manuscript by Feb 02 2022 11:59PM.   

Sincerely,

Beryne Odeny, 

PLOS Medicine

plosmedicine.org

Requests from Editors:

1. Please ensure that the updated title is submitted in the next version

2. Abstract: minor edit at line #77 “The limitation … was” 

3. Author summary: please restructure this section such that there are 2 to 4 bullet points per subheading. Each bullet point should not exceed 4 sentences

4. Please remove the “Role of funding source” from the methods section as this will be published based on your responses in the submission form

5. Conclusion: minor edit at line #494, should read “… products used for treatment fail …”

6. References – please provide weblinks and access dates for ref #8, 13, 14, 18, & 19

7. Please attach your STROBE checklists and any related flow diagrams.

Comments from Reviewers:

Reviewer #2: I reviewed verison R1 of this work. I have now looked at the revised manuscript and like to confirm that the comments have been, largely, addressed. In particular they have addressed the following issues: the study design, sample size calculation, randomization, missing cobalamin results, other infections, study profile, etc. 

Therefore I am be happy to recommend publication

Reviewer #4: The authors have addressed all my points.

Michael Dewey

[LINK]

---

## [Editor Report · Decision Letter 3]

3 Feb 2022

Dear Dr. Friis,

Thank you very much for re-submitting your manuscript "Serum cobalamin in children with moderate acute malnutrition in Burkina Faso: secondary analysis of a randomized trial" (PMEDICINE-D-21-02962R3) for review by PLOS Medicine.

I am pleased to say that provided the remaining editorial and production issues are dealt with we are planning to accept the paper for publication in the journal.

[LINK]

We look forward to receiving the revised manuscript by Feb 10 2022 11:59PM.   

Sincerely,

Beryne Odeny, 

PLOS Medicine

plosmedicine.org

Requests from Editors:

1) Thank you for providing your STROBE checklist. Please replace the page numbers with paragraph numbers per section (e.g. "Methods, paragraph 1"), since the page numbers of the final published paper may be different from the page numbers in the current manuscript.

2) The manuscript you have uploaded is not current as there are no observable edits. Please double check the version you uploaded and ensure you address prior editorial requests (no.2-6) as follows:

a. Main abstract: minor grammatical edit at line #77. It should read “The main limitation of this study was…” 

b. Author summary: please restructure this section such that there are 2 to 4 bullet points per subheading. 

c. Please remove the “Role of funding source” from the methods section as this will be published based on your responses in the submission form

d. Conclusion: minor grammatical edit at line #494. It should read “… products used for treatment fail …”

e. References – please provide weblinks and access dates for ref #8, 13, 14, 18, & 19

[LINK]

---

## [Editor Report · Decision Letter 4]

4 Feb 2022

Dear Dr. Friis,

Thank you very much for re-submitting your manuscript "Serum cobalamin in children with moderate acute malnutrition in Burkina Faso: secondary analysis of a randomized trial" (PMEDICINE-D-21-02962R4) for review by PLOS Medicine.

There is one more editorial remaining issue that needs to be addressed and it is listed at the end of this email. Please take this into account before resubmitting your manuscript:

[LINK]

We look forward to receiving the revised manuscript by Feb 11 2022 11:59PM.   

Sincerely,

Beryne Odeny, 

PLOS Medicine

plosmedicine.org

Requests from Editors:

1. Both STROBE and CONSORT checklists need revision. Please replace the page numbers with paragraph numbers per section (e.g. "Methods, paragraph 1"), since the page numbers of the final published paper may be different from the page numbers in the current manuscript

[LINK]

---

## [Editor Report · Decision Letter 5]

11 Feb 2022

Dear Dr Friis, 

On behalf of my colleagues and the Academic Editor, Prof. James K Tumwine, I am pleased to inform you that we have agreed to publish your manuscript "Serum cobalamin in children with moderate acute malnutrition in Burkina Faso: secondary analysis of a randomized trial" (PMEDICINE-D-21-02962R5) in PLOS Medicine.

PRESS

Sincerely, 

Beryne Odeny 

PLOS Medicine